# Digital Sustainable Entrepreneurship: A Digital Capability Perspective through Digital Innovation Orientation for Social and Environmental Value Creation

**Guangping Xu [1], Guangyuan Hou [2] and Jinshan Zhang [2,\*]**

1   School of Economics and Management, Changchun University of Science and Technology, Changchun 130022, China
2   School of Business and Management, Jilin University, Changchun 130012, China
\*   Correspondence: zhangjs@jlu.edu.cn

**Abstract:** With the rise of digital transformation in all domains, the relationship between digitalization and sustainable entrepreneurship has received growing attention. In practice, a new sustainable entrepreneurial model called "digital sustainable entrepreneurship" (DSE) has emerged. Aiming to establish a DSE model based on digital capability (DC) and digital innovation orientation (DIO), this study explored what kind of digital capability could be built to lead to a boost in digital sustainable entrepreneurship, to realize the creation of social and environmental value. We also revealed how DC affected DSE by introducing DIO and discussed the moderating role played therein by the manager's cognition of sustainable opportunities (MCSO). The study adopted CFA and SEM on the model using AMOS 27.0 and used the multiple regression analysis method to conduct an empirical study into the data from 308 SMEs in pollutive industries to validate the research framework. The results show a positive relationship between DC and DSE; DC is positively correlated to DIO; DIO is positively correlated to DSE; DIO plays a partial mediating role between DC and DSE; and MCSO positively moderates the relationship among DC, DIO, and DSE. This study will be of practical significance regarding how sustainable entrepreneurs can boost digital sustainable entrepreneurship.

**Keywords:** digital sustainable entrepreneurship; digital capability; digital innovation orientation; managers' cognition of sustainable opportunities; SMEs; pollutive industry



## 1. Introduction

Entrepreneurship has become a key solution for the sustainable development of the environment, resources, and community [1–3]. Despite the mainstream entrepreneurship research suggesting that entrepreneurs have a strong tendency towards self-interest and rational decision-making, seeking private economic incomes as the top consideration [4], entrepreneurs are always confronted with an ethical dilemma [5]. Traditional entrepreneurship research does not deny the positive effects of such idiosyncrasies contained in entrepreneurial spirits and activities as preserving the novelty, creativity, and sensitivity in the face of things upon addressing socioecological issues [6]. Some of the key actors who are called sustainable entrepreneurs take advantage of the entrepreneurial opportunities arising in the neglected social and environmental issues to create value [3]. With the business organization as a carrier, they attempt to shape consumers' value proposition by implementing the financial feasibility and an innovative business model; they enhance the socioecological education among youngsters and carry out indirect initiatives for biodiversity protection all throughout the society [7] to implement the entrepreneurial activities that give simultaneous consideration to the economic, environmental, and social benefits [3]. However, sustainable entrepreneurship is always confronted with the dual paradox that it is impossible to coordinate economic benefits and socioecological benefits. As a result, an organizations' sustainable entrepreneurship activity is widely low,

the continuity is weak, and the adopted proactive environmental strategy has difficulty persisting when the organization is facing turbulence in the business environment [8,9]. Some scholars' research has found that the entrepreneurial activities with a high level of innovation and institutional quality have a positive impact on sustainability [10]. The popularization of digital transformation has endowed corporate entrepreneurship and for the strategy of sustainable development with an innovative kernel [11], especially for SMEs with difficult sustainable practices and their own flexibility, the impact is most profound. Accordingly, tremendous changes to the rules of sustainable entrepreneurship within the SMEs have occurred, calling for the emergence of specialized theories on digital sustainable entrepreneurship [9,12,13].

Digital technology can be embedded into the existing technologies, products, and services [14]. Its features of analyticity, connectivity, perceptibility, and traceability not only enable the monitoring and evaluation of the pollutant discharges, waste recycling, and other sustainable destructive conducts of enterprises, but also can deliver the socioe-cological value proposition to consumers in a preferable manner, so that the neglected socially disadvantaged group can be involved in product design and value co-creation activities [15]. Particularly, since the outbreak of COVID-19, more attention has been paid to the innovative approaches of digitizing for addressing socioecological issues. This is because digital attributes are less restricted to geographic space and demonstrate excellent expandability; they can disarm the trade-off between profits and socioecological objec-tives, create a higher socioecological effect, and provide more feasible solutions to grand challenges [9]. Nowadays, the rising digital technology is playing a vital supportive role in the execution of the sustainable development goals (SDGs) of the United Nations [16]. The Chinese government's initiative for the development of digital economy and for the improvement in digital infrastructures has provided a vast prospect for the research on digital sustainable entrepreneurship. However, it needs to be considered that the digital technology in digital transformation cannot completely explain the strategic behavior that enterprises continuously carry out sustainable entrepreneurship. While the digital technol-ogy makes it possible to support the value proposition of environment–society–economy integration [9,17], its adoption has also introduced new sustainability issues, such as the intensive application of digital products along with the exponentially growing data size, the exploitation and use concerning rare metals, the recycling and treatment of wastes, and the considerable dissipation of electric power [9]. There seems to exist a natural contradiction between the digital technology and the environment [18].

According to the capability-based viewpoint, the different capabilities possessed by enterprises are an important source of the differences in behavior and performance between enterprises [19,20]. In the digital context, digital capability determines the frequency and outcome of enterprises' digitalization activities. Still, the capability reflects the completion of a special task, usually playing its role in a combined form and utilizing the organization flow to achieve the result [21]. Over recent years, the digital technology capability and digital dynamic capability proposed by scholars have transcended over the level of tech-nological application, providing a new perspective for studying the driving mechanism of digital sustainable entrepreneurship [22,23]. However, although digital capability has provided opportunities and convenience for organizations to access and implement digital sustainability, enterprises still need to keep digital innovation orientation to realize long-term digital sustainable entrepreneurship. The existence of digital innovation orientation provides a valuable perspective for digital initiatives within the organization [24]. Digital innovation will cause effective transformation and integration amid the acquired digital technology and the identified opportunities. Moreover, it will be used to guide the digital innovation practice per se and solve the sustainability conundrum, thereby unleashing the potential of digital technology to the maximum extent [25,26]. Furthermore, manager's cognition is an important reference for corporate behavior decision-making analysis, which is nonnegligible in corporate sustainable strategic initiatives [27]. According to the organi-zation structuration theory, organizer's cognition and action interdepend and interplay [8].

In the situation where enterprises widely carry out digital transformation and corporate digital entrepreneurship, only when managers hold a positive attitude towards socioecological issues will it be possible to input digital capability and digital innovation advantages into the campaigns that create value for the society and the environment. Unfortunately, the integration of sustainability and digitalization demand starts only where the private and public sectors obtain the preliminary practice [28], far from stimulating systematic and rigorous academic research, and lacking in corresponding empirical support.

This study aims to identify the composition of DC and its links to DSE. This study also attempts to uncover the mechanisms of this process and its influencing factors. The arrangement of the paper is as follows: Section 2 presents the literature progress on DSE and DC. Section 3 describes the theoretical model and assumptions of this study. The next section details the research methodology, Section 5 presents the test results of this study, Section 6 provides a discussion, and Section 7 includes the conclusions, contributions, managerial implications, and limitations and future research of this study.

## 2. Literature Review

### 2.1. Digital Sustainable Entrepreneurship

Sustainable entrepreneurship can be interpreted as something focused on the preservation of nature, life support, and community in the pursuit of perceived opportunities to bring into existence future products, processes, and services, where the pursuit of opportunities brings about the gain which is broadly construed to include economic and noneconomic gains to individuals, the economy, and society [3,29]. With the rise of digital transformation in all walks of life over recent years, the digitalization of enterprises brings unique changes to business operations, business processes, and cooperation methods [30,31]. Digital transformation is defined as an organizational shift towards big data, business analytics, cloud computing, mobility, and social media platforms [32]. It is not only a dual structure process composed of behavioral subjects and digital technology [33], but also a process of re-cognition and re-adaptation of strategy and structure of enterprises based on digital technology [34]. Whether from a strategic or organizational perspective, the beginning of digitalization means the beginning of a new model. Digital transformation can subvert traditional industries under the development of digital opportunities. Its essence is the process of corporate entrepreneurship or strategic renewal, and DC is one of the main outcomes of digital transformation [35]. From the perspective of digital transformation, Nambisan (2017) [11] pointed out that DC can be used to create new business models. Therefore, the more complete the DC system, the higher the corresponding entrepreneurial activity.

Si et al. (2020) [12] found that a growing number of inclusive entrepreneurships of embedded digital technology were deemed as an effective solution for poverty mitigation and thus for social inequality reduction. Some scholars have pointed out that the problems confronting sustainable entrepreneurship arise typically from the information asymmetry, which is manifested in the accuracy and timeliness of information transmission [9]. The analytical capability, connective capability, and intelligent capability gained with the application of diverse digital technologies by various enterprises can crack the market failure among economic, social, and environmental domains, thereby realizing the net positive environmental impacts [36,37]. Baranauskas and Raišienė (2021) [38] combined the Triple Bottom Line (TBL) method to build a conceptual framework of digital sustainable value creation, arguing that digital sustainable entrepreneurship represents the process of embedding social, environmental, and financial objectives into digital products, platforms, or ecosystems to realize sustainable value creation. George et al. (2021) [9] defined digital sustainable entrepreneurship as the organizational activities that seek the sustainable objective of boosting social and environmental value creation by creatively deploying and utilizing digital technology.

Nevertheless, the current digital sustainability remains a rising research field, where problems exist such as vagueness in theoretical conceptualization, overlap of terms, and

insufficient empirical evidence, lacking in a coherent analytical framework [9,38]. Table 1 displays some representative studies on digital sustainable entrepreneurship over recent years. Digital technology as a main influencing factor has been incorporated into the research system on the sustainable entrepreneurial process. At present, however, few scholars have reflected integrally on the relationship between corporate digitalization and corporate sustainable entrepreneurship and carried out the corresponding empirical study. Further exploration remains needed with respect to the formation mechanism of digital sustainable entrepreneurship.

**Table 1.** Studies on digital sustainable entrepreneurship (source: summarized by the author, 2022).

| Sources | Topic | Methods | Implication | Year |
|---------|-------|---------|-------------|------|
| [37] | How sustainable entrepreneurs embed digital technology into organizations' business models to promote social and environmental value creation | Qualitative research | Sustainable value proposition can be achieved ultimately by making selective use of digital technology in the business model | 2020 |
| [9] | How digital technology helps cope with the significant challenges of climate change and sustainable development | Theoretical research | Raised the management problems confronting sustainability and pointed out that digital sustainability could promote the progress in entrepreneurship, innovation, and strategy | 2021 |
| [13] | How the digital platform ecosystem creates social and environmental value | Synthesizing conceptual approach | Proposed a novel conceptual model of digital platform ecosystem, serving as a living laboratory for sustainable entrepreneurship and innovation | 2021 |
| [38] | Construction of a conceptual framework of digital sustainable value cycle | Theoretical research | A new interrelationship exists among digital entrepreneurship, sustainability, and business model | 2022 |
| [39] | The interrelationship between digital entrepreneurship and productive and innovative entrepreneurships and its impacts on EU countries attaining SDGs | Empirical research | National degree of digitalization positively affects the attainment of sustainable development goals | 2022 |

### 2.2. Digital Capability

In previous studies, scholars interpreted digital capability as a brand new competitiveness empowered to artificial intelligence, big data, cloud computing, and other rising digital infrastructures [40], while in the latest studies digital capability is defined as a set that ensures the transformation and integration of technological resources and that makes full use of these technological resources [41], including the analytical capability, connective capability, and intelligent capability and even highlighting the management efficiency of digital technology and the use of functions [29]. Specifically, digital capability is the starting point of corporate digital transformation since it integrates the advantages of digital technologies and digital professionals; it is the capability with which enterprises need to manage and make good use of digital technologies in the innovation process. Over recent years, some scholars have also been concerned regarding the digital dynamic capability. According to the dynamic capabilities theory, dynamic capabilities are the source from which enterprises gain sustainable competitive advantages. Recent studies consider the digital capability as a kind of dynamic capability [30]. Digital dynamic capability is defined as enterprises' responsiveness to changes in the market environment through restructuring digital technologies and creating new digital products and production flows, including three capabilities: digital perception, digital capture, and digital conversion [24]. The current research on digital capability is mostly focused on the relationship between digital capability and corporate digital transformation, as well as on the impacts of dig-

ital capability upon organizational performance and digital innovation [22]. Still, some scholars are concentrated on how to build the digital capability, underscoring the diversity of digital technologies [32] while overlooking the research on digital dynamic capability, and dissevering the association between digital technology operation and digital dynamic capability. Because of the existence of digital dynamic capability, digital capability can continually update digital resources and provide new opportunities for digital sustainable entrepreneurship. Digital technology also provides the condition for the formation and evolution of digital dynamic capability. Therefore, this paper interprets digital capability as the capability of an organization to utilize digital technology, digital knowledge, ideas, and updates to address enterprise operation and new business development, including digital technology capability and digital dynamic capability. Figure 1 displays the technological base, connotation system, and possible outcomes that constitute a digital capability.

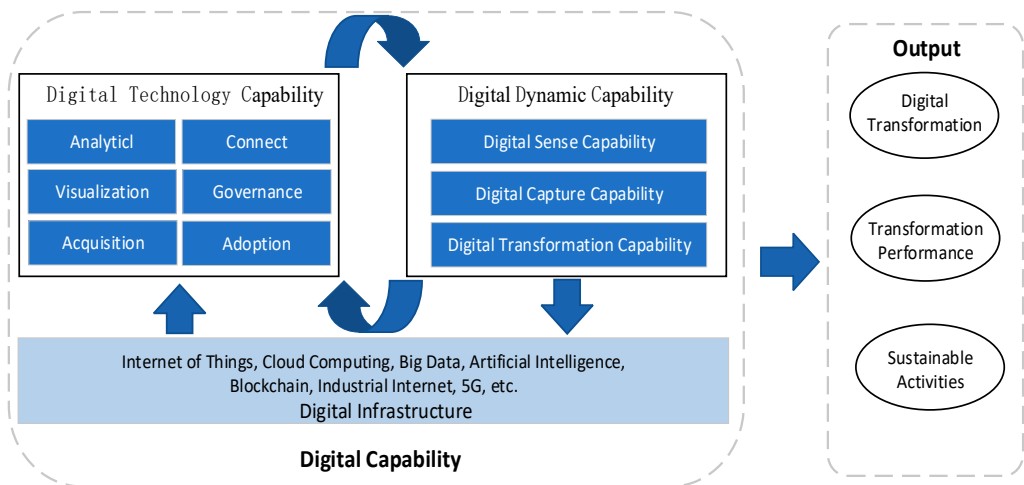

**Figure 1.** Digital technology and the digital dynamic-centered model of digital capability.

## 3. Theoretical Framework and Hypotheses

### 3.1. Digital Capability and Digital Sustainable Entrepreneurship

Many previous studies have discussed the positive impacts of digitalization on the innovation model and entrepreneurial behavior [28,32,35]. Nowadays, scholars widely believe that the success of organizations depends on the digital capability of grasping digital technologies and reining the digital world [19,35]. Digital capability reflects organizations' ability to identify, select, and update promising digital technologies to meet and fit specific demands, which converts the abstract digital technologies into specific digital solutions [41], providing internal conditions for addressing greenhouse emission, waste recycling, social inequality, poverty, and other equilibrium issues among nature, society, and economic activities [9]. SMEs in China's pollutive industries are subject to the strictest environmental and social supervision [8,42], and generally lack the capital R&D and investment advantages of large enterprises. They need to use digital intelligence and agility to respond to sustainable challenges. The studies have found that digital capabilities will facilitate the exchange of new ideas between SMEs and their value chain partners, which in turn will facilitate further improvement or optimization of business processes and related products [43]. This means that digital capabilities can reduce unnecessary material, energy, and other losses in the production process through lean management, reduce carbon emissions, help carbon peak and carbon neutrality, and help promote digital, networked, and intelligent production and achieve green energy-saving production, which is to realize social value [15].

Therein, digital technology capability addresses the barriers in sustainable activities to sustainable development, i.e., knowing, valuation, communication, and coordination by unleashing the advantages of digital technology, i.e., intelligence, analyticity, connectivity, eco-network property, and visuality [9]. The editability, re-programmability, functional

delay, and other features of digital technology provide a base for the development of digital dynamic capability [44]. Enterprises possessing digital dynamic capability are capable of swiftly upgrading or iterating digital infrastructures, responding to socioecological changes confronting sustainable development, and coping in time with changeable external social and environmental requirements. For instance, they are capable of coping readily with the transactional cost and ethical risk issues in the supply chain and maintaining the dominant position of sustainable products in an open market [9]. More importantly, via the rapid identification, capture, and reconstruction capabilities among digital dynamic capability, such enterprises are capable of piecing together sustainable entrepreneurial opportunities and resources in a creative manner and minimizing the impacts of the high energy consumption, radiation, and rare resource consumption on the society and environment brought about with intelligent equipment. Relying on the resources and capabilities of existing organizations, digital dynamic capability applies various digital technologies crosswise, improves the sustainable value creation potentials of existing services and products, and dynamically adjusts the sustainable business model, to ultimately promote the attainment of social and environmental value goals in digital sustainability [37,45]. According to the above analysis, this paper proposes the following hypothesis:

**Hypothesis 1 (H1).** *DC positively affects DSE.*

### 3.2. Digital Capability and Digital Innovation Orientation

Digital innovation refers to the creation of new digital products, the improvement in productive processes, the transformation of organizational and business models, and so forth, through adoption of a combination of information, computing, communication, and connectivity technologies in the innovation process [11,46]. The core of digital innovation remains the creation of existing value and new value [47]. Digital innovation orientation reflects the behavioral tendency of corporate digital activities, namely the digital innovation trend with high efficiency, novelty, and integrative nature arising from the digital transformation in products, services, and architectures for enterprises through utilization and integration of rising digital technologies [24]. Relying on digital technology's advantages in integrating and mobilizing resources, DC generates DIO, giving play to its supporting and driving roles in the innovation process, breaking the limit of the innovation stage, so that innovation exhibits the nonlinear development characteristics [11]. The digital technology capability is the guarantee for enterprises to implement digital innovations including product/service innovation, innovation of the business model, and innovation of the organizational structure [48], while the digital dynamic capability can help enterprises achieve continuous updates in digital technologies, resources, and capabilities. Organizations progressively develop the environmental and resource conditions for carrying out digital innovations while cultivating their digital technology capability, and the digital dynamic capability provides a spontaneous, incremental, thorough impetus for enterprises' innovation and transformation [24]. When organizations are more inclined towards resorting to digital capability for creating new approaches for business process and new business development, the level of sustainable digital innovation is raised in the organizational domain [49]. Therefore, this paper proposes the following hypothesis:

**Hypothesis 2 (H2).** *DC positively affects DIO.*

### 3.3. The Mediating Role of Digital Innovation Orientation

Digital capability is conducive to addressing the information asymmetry, data insufficiency, and accurate prediction ability shortage in digital sustainable entrepreneurship [9]. It can access, integrate, and utilize heterogeneous digital resources to create technological conditions for the development in sustainable businesses [39]. However, the digital capability relying on digital technology is not always a panacea for sustainable development. The latent contradictions between digital technology and energy and resources require

enterprises to keep a digital innovation orientation which represents the mechanism that involves the organization's creative behavior for the implementation of new methods in order to solve the existing problems via the emerging technologies [50], and remain proactive in the use efficiency, degree of novelty, and convergence effect of digital technology [24]. Both digital technology capabilities and digital dynamic capabilities are formed for innovation and development, and the ultimate goal of both is to serve the strategic results of enterprises [44]. Digital sustainable entrepreneurship, as the result of sustainable strategic actions under digital transformation, requires continuous local digital innovation processes to provide specific solutions. The digital sustainable entrepreneurial activities of enterprises in pollutive industries are confronted with the dynamic changes in demands, policies, and extreme environments. When the existing digital capability of enterprises is unable to address the barriers encountered in sustainable development, enterprises would unleash the flexibility, openness, and availability in the digital platform and the ecosystem induced over digital technology facilities via digital capability, to serve corporate innovation activities and rapidly assimilate, convert, and reconstruct the digital resources base with sustainable development potentials [51]. On the one hand, digital technologies creatively used via digital innovation orientation to attain the goals of reducing the costs on all parts of enterprise value chain and boosting the efficiency, including incorporating the operational philosophy of sustainable innovation into the design, R&D, production, marketing, and after-sales service links [52]. On the other hand, by proficiently unleashing the advantages of embedding digital technologies in, for example, homogeneity, re-programmability, and associativity, digital innovation orientation introduces new elements of value creation into enterprises, including products, services, architectures, and business models, to alleviate or radically resolve the contradiction between digitalization and sustainability [25]. The approaches of innovation calibration, balance, and blended value for digital innovation experiments provide a solution for enterprises' process of digital sustainable new business development to the problem that sustainable products are devoid of financial value. For instance, through the innovation in blockchain technology, a mechanism of consensus is established to implement untrusted exchanges, address the issue of governance failure, and expand the socioecological margin. In other words, DIO effectively associates DC and DSE, further unleashes the potential value in DC for digital technologies, digital opportunities, digital resources exploitation by deepening digital technology outcomes and innovation schemes, thereby achieving the compatibility among economic, social, and environmental value. According to the above analysis, this paper proposes the following hypotheses:

**Hypothesis 3 (H3).** *DIO positively affects DSE.*

**Hypothesis 4 (H4).** *DIO acts a mediating role between DC and DSE.*

### 3.4. The Moderating Role of Manager's Cognition of Sustainable Opportunities

Digitalization has brought about unprecedented opportunities and challenges to the environmental, social, and economic systems [53]. At the organization's microlevel, it becomes vitally important for enterprises, especially those in heavily pollutive industries, to tackle these challenges by adhering to a sustainability-prioritized and future-oriented attitude. From the resource-based view, resources and capabilities are the foundation affecting the direction and effectiveness of organizational action [54]. Management cognition is deemed as a resource unique to enterprises, which can increase or decrease corporate value [55], significantly affecting the retention of corporate activity advantages [56]. According to the theories of cognition, the manager can interpret or judge external environments or events in different ways by his/her own cognitive pattern; different cognitive patterns can influence the manager's behavior regarding different strategic options [57]. Previous studies have found that manager's environmental cognition, interpretation of opportunities in the natural environment, environmental commitment of senior management [58], and institutional environmental support [59] encourage enterprises to adopt forward-looking

environmental strategies; the difference in manager's cognition of social and environmental issues will make a difference to the sustainable strategies. Some studies have found that the manager's environmental cognition (e.g., belief, attitude, values, etc.) has a significant influence on corporate environmental protection and forward-looking environmental strategies [8,60]. This means when the manager occupied in digitalization strategies cognizes social and environmental issues as an opportunity for the enterprise, the enterprise will input more digital technology resources into social and environmental protection, and the corporate advantage in digital capability will be unleashed particularly with the delivered products and services to realize the social and environmental protection. Conversely, if the importance of sustainability is overlooked, then the enterprise's digital capability will serve the goal of business maximization, resulting in inordinate energy dissipation of digital technology, and increasing the social and environmental issues such as carbon footprints and information loss.

The research by Merrill et al. (2019) [28] found a very high failure rate of digital capability converting into sustainability, and that the failure was mainly due to the limitation in the conversion thinking. While studying how entrepreneurs utilized digital technology to design sustainable business models, Gregori (2020) [9] stressed the importance that managers associate digital logic with sustainable logic, noting that managers needed to discover the widely existing complementarity between digital capability and sustainable value creation. The manager's attitude towards social and environmental value could affect the effort level of the enterprise engaged in sustainable innovation activities; whether the enterprise can proactively implement environmental protection and social value creation depends on whether the manager interprets a sustainability issue as a market opportunity or as a social and environmental threat. The manager's cognition of sustainable opportunities can arouse individual sensitivity to positive outcomes, and it is also a process of the enterprise gathering sustainable knowledge and experience [61]. Once the unfulfilled demands in digital sustainability are found, decision-makers and first-tier managers can provide a clear goal for the design of the innovation prototype and boost the efficiency of converting digital technology capability and digital dynamic capability into digital innovation products, services, or business models, thereby improving the outcomes of digital innovation. Furthermore, the manager's interpretation of sustainable opportunities can help the enterprise acquire knowledge faster from the external society and environment, focus its vision on a farther future, guide digital innovation to develop and update towards digital sustainability, and promote the good match between digital innovation and digital sustainable entrepreneurship. Therefore, based on the above analysis, this paper proposes the following hypotheses:

**Hypothesis 5 (H5a).** *MCSO positively moderates the relationship between DC and DSE.*

**Hypothesis 5 (H5b).** *MCSO positively moderates the relationship between DC and DIO.*

**Hypothesis 5 (H5c).** *MCSO positively moderates the relationship between DIO and DSE.*

On the basis of theoretical research, we proposed a research model as shown in Figure 2. We examined how digital capability affects digital sustainable entrepreneurship, as well as the research problem that cognitive factors urge digital capability to enhance digital sustainable entrepreneurship.

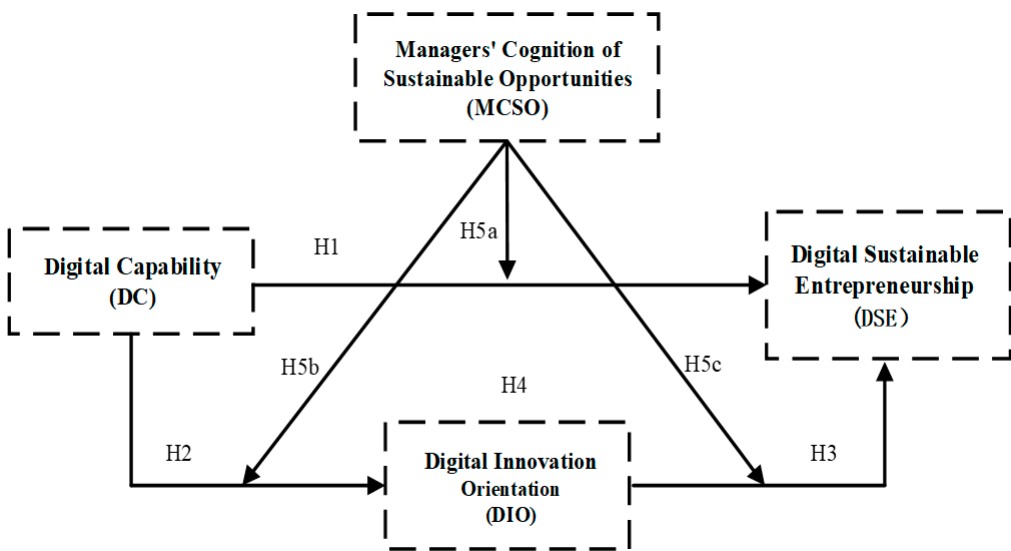

**Figure 2.** Research hypothesis.

## 4. Methodology

### 4.1. Sample and Data

Sustainability has become a common management principle pursued by the vast majority of enterprises [62]. The research topic of this paper is DSE. Digitization is the premise of the sample selection for this study. The level of digital transformation activities of SMEs in pollutive industries varies significantly. Not only do these SMEs have an important impact on the environment and society, they are more likely to receive pressured from the government and public to take environmental protection measures and implement sustainable entrepreneurship strategies. Selecting SMEs in the pollutive industry as our research sample is more conducive to revealing the process of DSE from the perspective of DC.

The samples of this study are from these SMEs, covering industries such as mining, papermaking, and textiles, which are representative to a certain extent. We take pollutive SMEs with industry specificity as samples to examine the laws of digital sustainable entrepreneurial activities. Other SMEs in non-polluting industries with general characteristics are also faced with social and environmental externalities. These SMEs also carry out digital transformation to different degrees and have different DCs. They also have positive references for the digital sustainable activities of SMEs in other industries.

The investigation object of this research group is the middle–senior managers (CEO, general/environmental department managers, etc.) of SMEs in pollutive industries. A senior or middle manager is selected in each SME, who plays an important decision-making role in the formulation, selection, and implementation of the enterprise sustainable strategy, and can reflect the overall sustainable cognition level of the organization. The division of SMEs is determined according to the industry and operating income stipulated in the "SMEs Promotion Law of the People's Republic of China", and the classification standards for different types of industries are different. For example, industrial SMEs generally have less than 1000 employees or an operating income of less than RMB 400 million. SMEs in pollutive industries are an important part of the national economy, these SMEs are widely confronted with environmental and social pressures, are undertaking digital transformation activities, and are more likely to take digitally sustainable entrepreneurial actions. Thus, samples were selected using stratified random by the standard of classifying enterprises in pollutive industries including printing, petroleum, electric power, and heating power determined in "Environmental Information Disclosure Intelligence of Listed Companies" published by the Ministry of Ecology and Environment of the People's Republic of China. In data collection, this study combined on-site and online questionnaire surveys. With the help of industry associations, 400 questionnaires were issued, 352 were returned and 44



that were invalid were ruled out to retrieve 308 valid questionnaires (effective reply rate: 77%). The detailed information about the sampled enterprises in this study is shown in Table 2.

**Table 2.** Sample specifics.

| Sample | Options | Sample Size | Percentage (%) | Sample | Options | Sample Size | Percentage (%) |
|---|---|---|---|---|---|---|---|
| Position | CEO/Chairman/Manager | 54 | 17.532 | Property | State owned | 125 | 40.584 |
| | Business entity | 37 | 12.013 | | Private | 183 | 59.416 |
| | R&D/Market/ Manufacture Manager | 107 | 34.74 | Income (average revenue in the last 3 years) | Less than 3 million | 60 | 19.481 |
| | | | | | 3–5 million | 104 | 33.766 |
| | Environment/Health/ Security Manger | 110 | 35.714 | | 5–8 million | 102 | 33.117 |
| Gender | Male | 193 | 62.662 | | 8–10 million | 42 | 13.636 |
| | Female | 115 | 37.338 | | | | |
| Education level | Undergraduate | 49 | 15.909 | | | | |
| | Postgraduate | 68 | 22.078 | | | | |
| | MBA/EMBA | 101 | 32.792 | | | | |
| | PhD | 90 | 29.221 | Industry | Mining | 7 | 2.273 |
| Age | Under 25 | 47 | 15.26 | | Food/ Beverage | 64 | 20.779 |
| | 25–35 | 64 | 20.779 | | Textile/Clothing/ Leather | 61 | 19.805 |
| | 36–45 | 94 | 30.519 | | Paper/Printing | 74 | 24.026 |
| | Over 46 | 103 | 33.442 | | Oil/Chemistry/Plastic | 10 | 3.247 |
| Number of employees | Under 50 | 37 | 12.013 | | Metal/Non-metallic | 12 | 3.896 |
| | 51–100 | 32 | 10.39 | | Machinery/Facility/ Instrument | 10 | 3.247 |
| | 101–200 | 83 | 26.948 | | Pharmacy/Biology | 50 | 16.234 |
| | 201–300 | 79 | 25.649 | | Electric/Heat/Water | 10 | 3.247 |
| | 301–1000 | 77 | 25 | | Tobacco | 10 | 3.247 |

### 4.2. Measurement of Variables

Combining relevant domestic and overseas studies and the opinions of industrial informatization experts, local governmental personnel, and researchers, this study adopted well-developed scales to measure the main variables. These indexes were described using Likert 7-point scales. The concrete measures are shown in Table A1.

DSE. DSE is an organizational activity that seeks to boost the sustainable development goals by creatively deploying and utilizing digital technologies. Taking example from digital entrepreneurship and sustainable entrepreneurship, this paper draws on the measurement scales of Shepherd et al. (2011) [62] and Baranauskas and Raišienė (2022) [38] on digital entrepreneurship and sustainable entrepreneurship and measures digital sustainable entrepreneurship using 6 items. Sample questions include "Strip several businesses that go against the integration of digital technology with social and environmental value creation" (Cronbach's α = 0.887).

DC. Capability usually plays its role in a combined form. The digital capability studied in this paper falls into two dimensions: digital technology capability and digital dynamic capability. In total, 11 items from Zhou et al. (2010) [63] and Annarelli (2021) [40] were adopted to measure DC, 3 for digital technology capability, sample questions including "acquire important digital technologies", and 8 for digital dynamic capability, sample questions including "identify new digital opportunities" (Cronbach's α = 0.903).

DIO. This paper expands the research in three dimensions: efficiency, novelty, and convergence, adopting nine items from Von (2018) [64] to measure DIO, three for the dimension "efficiency", sampling questions including "We use digital technology to deliver new offerings as a goal" (Cronbach's $\alpha$ = 0.808).

MCSO. MCSO can arouse employees' sensitivity to positive outcomes, and it is also a process of the enterprise gathering sustainable knowledge and experience. This paper takes example from four items from White et al. (2003) [65] and Liu et al. (2013) [66] to measure MCSO, sample questions including "For the development of this company, I think the confronting natural and social environments are positive" (Cronbach's $\alpha$ = 0.869).

### 4.3. Control Variables

Manager characteristics and enterprise characteristics are key factors affecting strategic behaviors. To ensure the rigorousness of the study, this paper takes example from the study by Sharma (2000) [67], selecting manager characteristics (manager's position, gender, and educational background) and enterprise characteristics (number of years since the foundation of the enterprise, number of employees, nature of property right, average input over the last three years, and industry involved) as the control variables affecting DSE.

### 4.4. Confirmatory Factor Analysis (CFA)

To assure that the scales of all variables can effectively capture the corresponding variables, a CFA was conducted on the main variables: DC, DIO, DSE, and MCSO. The results of the CFA using Amos27.0 are shown in Table 3. When the seven-factor model was adopted, the fitting result was better than other models ($\chi^2$ = 519.224; Df = 384; TLI = 0.970; CFI = 0.974; RMR = 0.137; RMSEA = 0.034), showing that the variables designed in this paper have good discriminant validity.

**Table 3.** CFA results.

| Model | Fit Indices | Model | Fit Indices |
|---|---|---|---|
| Seven factors DC-A, DC-B, DIO-A, DIO-B, DIO-C, DSC, MCSO | $\chi^2$/df = 1.352 TLI = 0.970 CFI = 0.974 RMR = 0.137 RMSEA = 0.034 | Six factors DC-A + DC-B, DIO-A, DIO-B, DIO-C, DSC, MCSO | $\chi^2$/df = 2.099 TLI = 0.907 CFI = 0.974 RMR = 0.137 RMSEA = 0.034 |
| Six factors DC-A, DC-B, DIO-A + DIO-B, DIO-C, DSC, MCSO | $\chi^2$/df = 2.150 TLI = 0.902 CFI = 0.912 RMR = 0.207 RMSEA = 0.061 | Six factors DC-A, DC-B, DIO-A, DIO-B + DIO-C, DSC, MCSO | $\chi^2$/df = 2.098 TLI = 0.907 CFI = 0.916 RMR = 0.187 RMSEA = 0.06 |
| Five factors DC-A, DC-B, DIO-A + DIO-B, DIO-C, DSC, MCSO | $\chi^2$/df = 2.874 TLI = 0.841 CFI = 0.856 RMR = 0.246 RMSEA = 0.078 | Five factors DC-A + DC-B, DIO-A, DIO-B, DIO-C + DSC, MCSO | $\chi^2$/df = 2.821 TLI = 0.845 CFI = 0.86 RMR = 0.22 RMSEA = 0.077 |
| Five factors DC-A, DC-B, DIO-A + DIO-B, DIO-C + DSC, MCSO | $\chi^2$/df = 2.864 TLI = 0.842 CFI = 0.856 RMR = 0.232 RMSEA = 0.078 | Four factors DC-A + DC-B, DIO-A + DIO-B + DIO-C, DSC, MCSO | $\chi^2$/df = 3.581 TLI = 0.781 CFI = 0.799 RMR = 0.238 RMSEA = 0.092 |
| Three factors DC-A + DC-B, DIO-A + DIO-B + DIO-C, DSC + MCSO | $\chi^2$/df = 4.980 TLI = 0.662 CFI = 0.688 RMR = 0.294 RMSEA = 0.114 | Two factors DC-A + DC-B + DIO-A + DIO-B + DIO-C, DSC + MCSO | $\chi^2$/df = 6.217 TLI = 0.557 CFI = 0.589 RMR = 0.378 RMSEA = 0.130 |

**Table 3.** *Cont.*

| Model | Fit Indices | Model | Fit Indices |
|---|---|---|---|
| One factor<br>DC-A + DC-B + DIO-A +<br>DIO-B + DIO-C +<br>DSC + MCSO | $\chi^2/\text{df} = 7.191$<br>TLI = 0.474<br>CFI = 0.511<br>RMR = 0.345<br>RMSEA = 0.142 | | |

Notes: N = 308; DC-A: Digital Technology Capability; DC-B: Digital Dynamic Capability; DIO-A: Efficiency; DIO-B: Novelty; DIO-C: Convergence; DES: Digital Sustainable Entrepreneurship; MCSO: Manager's cognition of sustainable opportunities; + denotes that the factors are synthesized into one variable.

### 4.5. Common Method Variance

Against the potential issue of common method variance (CMV), this study tested the CMV using the Harman single factor method proposed by Podsakoff et al. (1986) [68]. The principal component analysis (PCA) method was employed on the whole questionnaire. A total of seven factors with eigenvalues greater than 1 were extracted in the nonrotation case, cumulatively accounting for 70.627% of the overall variation. The first factor accounted for 32.282% of the variance, with the dependent variable and the independent variable loading different factors. Hence, CMV is not serious. Moreover, CFA can also be used to test CMV. Our results also demonstrate that no homologous factor affecting the model estimation exists.

In the questionnaire measurement, the social desirability variance (SDV) received much attention. To reduce SDV, on the one hand, we highlighted the technicality of this questionnaire in large font size and black boldface on the homepage of the questionnaire; on the other hand, following the method of Banerjee (2001) [69], we promised that the research results would be aggregated, i.e., the respondents would not be identified individually, and that the questionnaire was involved with specific actions and strategic issues rather than pertaining to general moral requirements.

## 5. Results

### 5.1. Characteristics of Samples

SPSS Statistics version 21(IBM SPSS Inc., Chicago, IL, USA) was used for Pearson correlation coefficient analysis and the specific results are shown in Table 4.

**Table 4.** The descriptive analysis and correlation coefficients.

| | 1 | 2 | 3 | 4 | 5 | 6 | 7 | 8 | DC | DIO | MCSO | DSE |
|---|---|---|---|---|---|---|---|---|---|---|---|---|
| Position | 1 | | | | | | | | | | | |
| Gender | −0.149 ** | 1 | | | | | | | | | | |
| Education level | 0.260 ** | −0.146 * | 1 | | | | | | | | | |
| Age | 0.053 | −0.066 | 0.057 | 1 | | | | | | | | |
| Number of employees | 0.301 ** | −0.049 | 0.140 * | 0.132 * | 1 | | | | | | | |
| Property | 0.109 | −0.1 | 0.216 ** | 0.142 * | 0.202 ** | 1 | | | | | | |
| Income | −0.262 ** | 0.104 | −0.379 ** | −0.195 ** | −0.324 ** | −0.176 ** | 1 | | | | | |
| Industry | −0.203 ** | 0.051 | −0.278 ** | −0.184 ** | −0.105 | −0.165 ** | 0.105 | 1 | | | | |
| DC | −0.032 | 0.066 | −0.061 | −0.031 | 0.065 | −0.053 | −0.022 | 0.009 | 1 | | | |
| DIO | 0.063 | 0.069 | −0.014 | 0.004 | 0.078 | 0.03 | −0.104 | −0.064 | 0.411 ** | 1 | | |
| MCSO | 0.161 ** | −0.085 | −0.072 | −0.087 | 0.179 ** | 0.071 | −0.071 | −0.068 | 0.302 ** | 0.314 ** | 1 | |
| DSE | 0.03 | 0.083 | −0.002 | −0.105 | −0.023 | −0.095 | −0.046 | −0.003 | 0.589 ** | 0.605 ** | 0.279 ** | 1 |
| M | 2.886 | 1.373 | 2.753 | 2.821 | 3.412 | 1.594 | 3.347 | 3.555 | 4.093 | 4.174 | 4.129 | 4.052 |
| SD | 1.081 | 0.484 | 1.045 | 1.060 | 1.295 | 0.492 | 2.011 | 2.265 | 1.183 | 1.080 | 1.396 | 1.271 |

Note: (* $p < 0.05$, two tailed) (** $p < 0.01$, two tailed).

### 5.2. Hypothesis Testing

5.2.1. Results of Main and Mediating Effect Tests

We adopted the method of hierarchy regression analysis (HRA) to analyze the relationship between DC and DSE. From Table 5, Model 1-1 is the regression result of the control variable and DC with respect to DSE. Evidently, DC has a significant positive direct impact on DSE (Model 1: β = 0.631, SE = 0.05, $p < 0.01$). H1 is validated. Model 4 is the regression result of the control variable and DC with respect to DIO. Evidently, DC has a significant positive impact on DIO (Model 4: β = 0.370, SE = 0.048, $p < 0.01$). H2 is validated. This shows that, when possessing digital capability, enterprises may identify the digital sustainable opportunities, restructure the digital resources, expand the sustainable innovation level of products and services, and build a sustainable image by introducing the state-of-the-art digital technologies, thereby promoting corporate digital sustainable entrepreneurial activities for the better.

**Table 5.** Results of the test of direct and mediating effects (N = 308).

| Variable | DSE | | | DIO |
|---|---|---|---|---|
| | **Mode 1** | **Mode 2** | **Mode1 3** | **Mode 4** |
| Control variable | | | | |
| (Constant) | 1.948 ** | 1.671 | 0.565 | 2.663 ** |
| | (0.542) | (0.539) | (0.49) | (0.523) |
| Position | 0.077 | 0.018 | 0.047 | 0.058 |
| | (0.059) | (0.058) | (0.052) | (0.057) |
| Gender | 0.13 | 0.085 | 0.063 | 0.13 |
| | (0.123) | (0.121) | (0.107) | (0.119) |
| Education level | 0.03 | 0.045 | 0.061 | −0.06 |
| | (0.064) | (0.063) | (0.056) | (0.062) |
| Age | −0.103 | −0.107 | −0.094 | −0.015 |
| | (0.058) | (0.057) | (0.05) | (0.055) |
| Number of employees | −0.08 | −0.052 | −0.081 | 0.001 |
| | (0.05) | (0.049) | (0.043) | (0.048) |
| Property | −0.145 | −0.251 | −0.188 | 0.083 |
| | (0.125) | (0.123) | (0.109) | (0.121) |
| Income | −0.04 | −0.013 | −0.013 | −0.052 |
| | (0.034) | (0.033) | (0.029) | (0.032) |
| Industry | −0.01 | 0.007 | 0.005 | −0.029 |
| | (0.028) | (0.027) | (0.024) | (0.027) |
| Independent variable | | | | |
| DC | 0.631 ** | | 0.439 ** | 0.370 ** |
| | (0.05) | | (0.047) | (0.048) |
| DIO | | 0.716 ** | 0.519 ** | |
| | | (0.054) | (0.052) | |
| $R^2$ | 0.371 | 0.392 | 0.529 | 0.191 |
| Adjusted $R^2$ | 0.352 | 0.374 | 0.513 | 0.166 |
| F | 19.54 | 21.371 | 33.32 | 7.793 |
| VIF (max) | 0.73 | 1.352 | 1.352 | 1.322 |

Note: ** $p < 0.01$.

Regarding the test on the mediating role, this paper adopted the three-step method proposed by Baron and Kenny (1986) [70] to test the mediating role of DIO. From Model 2 in Table 5, it can be known that DIO has a significant positive impact on DSE (Model 2: β = 0.716, SE = 0.054, $p < 0.01$). H3 is validated. From Model 3, the effect of DC on DES remains significant, though the regression coefficient decreases, after adding the mediating variable DIO on the basis of Model 1, showing that DIO has a partial mediating role in the relationship between DC and DSE. H4 is tenable. To further confirm the mediating role of DIO, the nonparametric percentage Bootstrap method was adopted for testing, according to the suggestion of Hayes et al. (2013) [71]. While the mediating role of DIO between DC

and DSE was being tested, the sample size was set as 5000. From Table 6, the value of the indirect effect among the models is 0.192. The Boot 95% confidence interval is [0.139, 0.251], excluding 0, suggesting that the mediating role of DIO between DC and DSE is significant. Figure 3 shows the resulting measurement and path model.

**Table 6.** Results of bootstrapping analysis of mediation effects.

|  | Effect | SE | LLCI | ULCI |
|---|---|---|---|---|
| Total effect | 0.631 | 0.049 | 0.533 | 0.729 |
| Direct effect | 0.439 | 0.047 | 0.346 | 0.532 |
| Indirect effect | 0.192 | 0.028 | 0.139 | 1.251 |

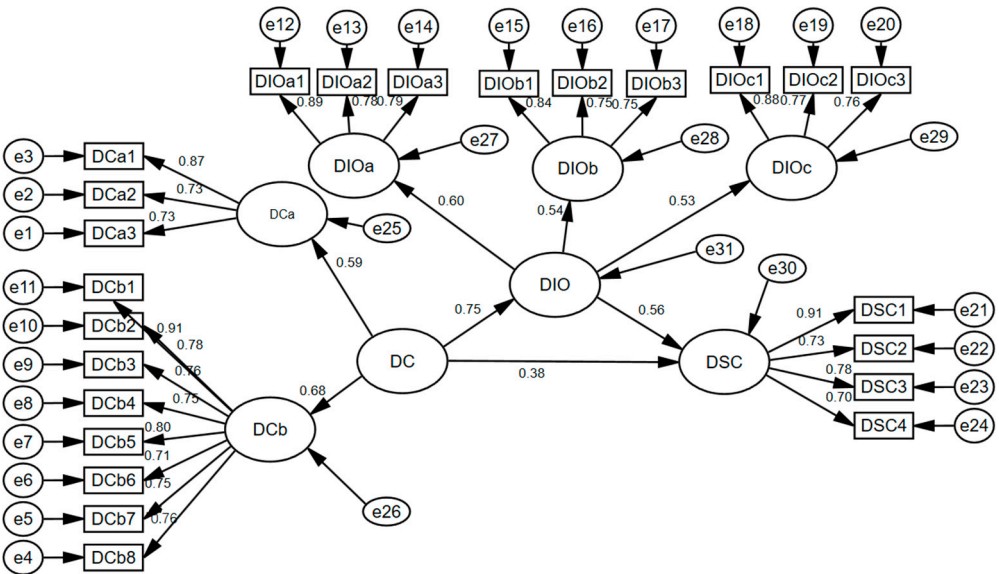

**Figure 3.** Measurement and path model.

5.2.2. Test of the Moderating Role

Table 7 displays the moderating effect of MCSO on the relationship among DC, DIO, and DSE. Model 4 is the regression result of the control variable and independent variable (DC), the moderating variable (MCSO), and the interaction item (MCSO × DC) with respect to the dependent variable (DSE) ($\beta = 0.199$, SE = 0.061, $p < 0.01$). DC, MCSO, and MCSO × DC all have a significant positive impact on DSE, that is, relative to a low cognition of sustainable opportunities, when the manager has a high cognition of sustainable opportunities, enterprises with higher DC are more likely to carry out DSE. Hypothesis 5a is validated. Model 5 is the regression result of the control variable and independent variable (DIO), the moderating variable (MCSO), and the interaction item (MCSO × DIO) with respect to the dependent variable (DSE) ($\beta = 0.080$, SE = 0.06, $p > 0.01$). Relative to a low cognition of sustainable opportunities, when the manager has a high cognition of sustainable opportunities, enterprises that maintain DIO are more likely to adopt DSE. Hypothesis 5c is validated. Model 6 is the regression result of the control variable and independent variable (DC), the moderating variable (MCSO), and the interaction item (MCSO × DC) with respect to the mediating variable (DIO) ($\beta = 0.182$, SE = 0.058, $p < 0.01$). DC, MCSO, and MCSO × DC all have a significant positive impact on DIO, that is, relative to a low cognition of sustainable opportunities, when the manager has a high cognition of sustainable opportunities, enterprises with higher DC are more likely to maintain a high level of DIO. Hypothesis 5b is validated. Furthermore, the maxima of VIF are all below 10, indicating that there is no serious multicollinearity with the models.

**Table 7.** Results of the test of the moderating effect.

| Variable | | | DSE | | | DIO |
|---|---|---|---|---|---|---|
| | **Mode 1** | **Mode 2** | **Mode 3** | **Mode 4** | **Mode 5** | **Mode 6** |
| Control variable | | | | | | |
| (Constant) | 4.76 (0.612) | 1.948 (0.542) | 1.582 (0.557) | 1.611 (0.549) | 1.283 (0.549) | 2.180 (0.524) |
| Position | 0.047 (0.074) | 0.077 (0.059) | 0.057 (0.060) | 0.058 (0.059) | 0.004 (0.058) | 0.032 (0.056) |
| Gender | 0.212 (0.152) | 0.13 (0.123) | 0.161 (0.123) | 0.148 (0.121) | 0.110 (0.120) | 0.160 (0.115) |
| Education level | −0.017 (0.079) | 0.03 (0.064) | 0.055 (0.065) | 0.073 (0.064) | 0.078 (0.063) | −0.008 (0.061) |
| Age | −0.13 (0.071) | −0.103 (0.058) | −0.083 (0.058) | −0.082 (0.057) | −0.078 (0.056) | 0.012 (0.054) |
| Number of employees | −0.03 (0.062) | −0.08 (0.05) | −0.094 (0.050) | −0.085 (0.049) | −0.065 (0.049) | −0.010 (0.047) |
| Property | −0.226 (0.155) | −0.145 (0.125) | −0.168 (0.125) | −0.193 (0.123) | −0.272 (0.121) | 0.028 (0.117) |
| Income | −0.058 (0.042) | −0.04 (0.034) | −0.036 (0.033) | −0.029 (0.033) | −0.005 (0.033) | −0.040 (0.031) |
| Industry | −0.017 (0.034) | −0.01 (0.028) | −0.004 (0.028) | −0.007 (0.027) | 0.008 (0.027) | −0.023 (0.026) |
| Independent variable | | | | | | |
| DC | | 0.631 ** (0.05) | 0.592 ** (0.052) | 0.559 ** (0.052) | | 0.285 ** (0.050) |
| DIO | | | | | 0.655 ** (0.056) | |
| Moderator | | | | | | |
| MCSO | | | 0.113 ** (0.046) | 0.114 ** (0.045) | 0.115 ** (0.045) | 0.159 ** (0.043) |
| Interaction | | | | | | |
| MCSO × DC | | | | 0.199 ** (0.061) | | 0.182 ** (0.058) |
| MCSO × DIO | | | | | 0.134 ** (0.058) | |
| $R^2$ | 0.033 | 0.371 | 0.384 | 0.405 | 0.388 | 0.405 |
| Adjusted $R^2$ | 0.007 | 0.352 | 0.363 | 0.383 | 0.365 | 0.383 |
| F | 1.261 | 19.54 | 18.506 | 18.348 | 17.028 | 18.348 |
| VIF (max) | 1.338 | 1.34 | 0.686 | 0.809 | 1.362 | 1.366 |

Note: ** $p < 0.01$.

To further verify the moderating role of MCSO in the relationship among DC, DIO, and DSE, we constructed the graphs of the moderating effect (see Figures 4–6) using the method of Aiken and West (1991) [72]. The results of a simple slopes test show that when MCSO is at a low level (one standard deviation below the mean) the impact of DC on DSE is weak (B = 0.463, Boot 95% CI [0.278, 0.648]); when MCSO is at a high level (one standard deviation above the mean), the impact of DC on DSE is strong (B = 0.860, Boot 95% CI [0.708, 1.012]). From Figure 4, compared with low MCSO, high MCSO enhances the positive relations.

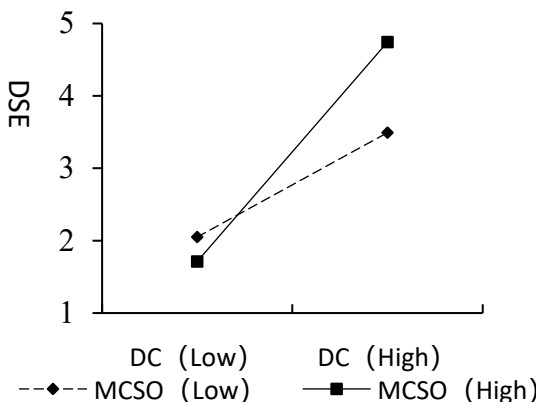

**Figure 4.** Interactive effects of DC and MCSO on DSE.

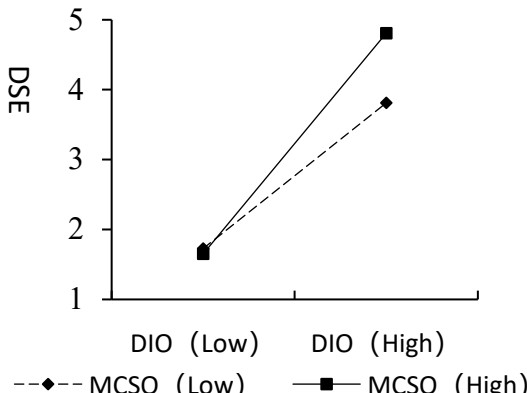

**Figure 5.** Interactive effects of DIO and MCSO on DES.

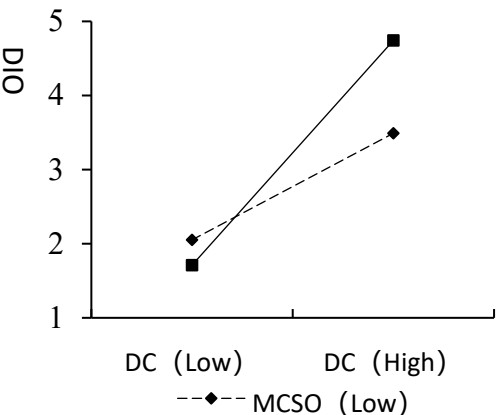

**Figure 6.** Interactive effects of DC and MCSO on DIO.

Figure 5 indicates that when MCSO is at a low level, the impact of DIO on DSE is weak (B = 0.573, Boot 95%CI [0.397, 0.749]); when MCSO is at a high level, the impact of DIO on DSE is strong (B = 0.841, Boot 95% CI [0.690, 0.992]). From Figure 5, compared with low MCSO, high MCSO enhances the positive relationship between DIO and DSE.

Figure 6 indicates that when MCSO is at a low level, DC has no significant impact on DIO (B = 0.156, Boot 95% CI [−0.020, 0.749]); when MCSO is at a high level, DC has a significant positive impact on DIO (B = 0.519, Boot 95% CI [0.374, 0.665]). From Figure 6, compared with low MCSO, high MCSO enhances the positive relationship between DC and DIO.

## 6. Discussion

The current social and environmental issues are widely embedded in the production and livelihood of human society. These sustainability issues need to be addressed in different domains, such as policy, economics, society, and enterprise. Meanwhile, the digital transformation in all domains triggered by digital technology has provided brand-new solutions to social and environmental sustainability issues. The relationship between corporate digital transformation and corporate sustainable development has always been ambiguous [18], and the research on DSE is extremely rarely found in the literature [9,70]. Our research suggests that the digital capability composed of digital technology capability and digital dynamic capability is capable of raising the level of socioecological value creation of digital sustainable entrepreneurial activities of these SMEs in pollutive industries, which agrees with the theoretical hypothesis proposed by Howard-Grenville et al. (2014) [2]. This means that the construction of the digital capability of pollutive SMEs occupied in digital transformation can indeed boost the synchronous creation of corporate economic, social, and environmental values. This conclusion echoes the proposal for the framework of digital sustainable entrepreneurial value raised by Baranauskas and Raišienė (2022) [38] and lends support to the conclusion drawn by Herman (2022) [39] that digital entrepreneurship can positively affect the realization of the national sustainable development goals. The study by Gregori et al. (2020) [9] also suggested that selective use of digital technologies enabled the realization of the balanced value proposition as per the need for sustainability while delivering conveniency and efficiency. We conducted a comprehensive analysis into the content system of digital capability from the perspective of capability, elaborating on how digital capability provided reliable implementation conditions for sustainable entrepreneurial practices of SMEs.

Furthermore, we found that DIO mediated the relationship between DC and DSE to some extent. On the one hand, digital capability raises the feasibility of enterprises carrying out digital innovation, providing innovation conditions for enterprises to continually introduce and assimilate digital technology advantages, which is beneficial for original, incremental, and radical innovation [66], and emissions and waste, improving supply chain management [73]. Having a high degree of DC enables employees to track business processes in real time, making workflows more transparent. This allows for identification of processes that can be digitally improved or enhanced [49]. On the other hand, DIO can aggregate various digital practices, influence and fulfill the need of sustainable development for multigoal coordination, and crack the resource contradiction between digitalization and sustainability [24]. Currently, the mainstream research supports the positive impact of digitization on sustainable activities [72,73], but there are also inconsistent views that digitization is "energy hungry" and resource-intensive, with negative impacts on the environment [74–76]. Ahmadova et al. (2022) combined the above viewpoints and took the national digitization level as the research object and believe that digitization has a positive impact on environmental performance, but it reaches a tipping point. Excessive digitization leads to a "rebound effect", thus increasing the use of resources and leading to higher pollution [18], while we supplement this phenomenon at the micro level. The DC of SMEs can continue to alleviate the contradiction between digitalization and the environment through DIO and continue to promote sustainable digital development. Our study further determined the factors influencing the activity of corporate digital sustainable entrepreneurship and, at the empirical level, discussed and substantiated the implementation process between digital entrepreneurship and sustainable development solutions within SMEs.

The cognition of sustainability with entrepreneurs boosting the change in dominant practices and mindsets guides the mutual promotion between corporate digital transformation and sustainable entrepreneurship. In this study, MCSO positively moderates the separate relationships between DC, DIO, and DSE. This conclusion agrees with the viewpoint raised by Sharma (2000) [67] and He et al. (2019) [42] and other scholars. The studies by some scholars found that the higher the manager's attention to social and envi-

ronmental issues is, the more likely the manager is to perceive sustainable development opportunities and access the relevant information [39], and the more ready the manager is to input the existing digital resources into sustainable entrepreneurial practice [77]. Based upon the expectancy theory, when the enterprise has integral digital capability, it has more confidence to utilize digital capability and digital innovation to address social and environmental issues under the guide of MCSO, implement the exploitation of social and environmental opportunities, and select the relatively advanced strategies over those of rivals [78]. The contingency theory suggests that the influences of enterprise resources and competence on organizational strategic behaviors are contingent on the internal and external environments in which the enterprise is situated [79]. Sabbir found that creating positive environmental attitudes and habits through workplace environmental policies, procedures, and practices can improve employees' task-related EGB [27]. Reflecting the interpretation of decision-makers about the external sustainable environment, MCSO can arouse the manager's positive emotion, relieve the manager's psychological discomfort reactions, and promote the manager's pursuit for the sustainable growth, development, cultivation, and other internal demands of digital transformation. Furthermore, our study also found that high MCSO could strengthen the relationship between DC and DIO, while low MCSO is unable to affect the relationship between both. This shows that high cognition of sustainable opportunities provides extensive sources for digital innovation and creates advantages for carrying out digital innovation. According to the above research findings, it can be deemed that sustainable entrepreneurship can play an important driving role in pursuing digital sustainability and provide a precondition for addressing many complex social and environmental issues. Unfortunately, the domain of digital sustainable entrepreneurship remains at the starting stage, without in-depth exploration into the formation mechanism digital sustainable entrepreneurship under the digital economy context. Our objective is to make contributions to the domain of digital sustainable entrepreneurship by testing an organizational ability and manager's cognitive factors boosting the digital sustainable development.

## 7. Conclusions

We tested a process model to study how digital capability acts on digital sustainable entrepreneurship of SMEs. During this process, the construction and promotion of digital capability can boost and facilitate digital sustainable entrepreneurship of SMEs. Digital innovation orientation acts a partial mediating role between digital capability and digital sustainable entrepreneurship. Digital capability reconciles potential contradictions via digital innovation as digital technology serves sustainable development of SMEs, delivering the effect of digital capability on social value creation. Furthermore, managers with the cognition of sustainable opportunities are more likely to transit digital capability and digital innovation orientation to digital sustainable entrepreneurship of social and environmental value creation, and they can also promote the positive effects of digital capability on digital innovation orientation.

### 7.1. Contribution to Research

First, this research enriched the influencing factors of digital sustainable entrepreneurship of SMEs and tested the relationship between DC and DSE. Under the concept of sustainable development, what kind of organizational ability can help the enterprise further boost sustainable entrepreneurial activities? Previous studies on sustainable entrepreneurship tend to explore the connotation, obstructive factors of sustainable entrepreneurship, as well as the mechanism of sustainable entrepreneurship and the business model design in the process of this activity, mostly from the Theory of Planned Behavior, the institutional theory, the stakeholder theory, and the strategic environment view. Overall, few studies discuss the social and environmental aspects of sustainable development from a capacity perspective [39]. The study by Gregori and Holzmann (2020) [80] suggests digital technology supports the development in the value proposition of integrating environmental,

social, and economic benefits. The digital capability based on digital infrastructures has become a new start to addressing sustainable development issues, but at present studies remain deficient on the mechanism of sustainable entrepreneurial activities under the perspective of digital capability. Distinguished from some scholars' emphasis on the functional application of digital technology [23,48], we demonstrated whether the digital capability composed of the two representative capabilities (digital technology capability and digital dynamic capability) can propel digital sustainable entrepreneurial activities of SMEs on the basis of analyzing the constitution foundation of digital capability. Having brought digital dynamic capability into the scope of analysis, this paper allowed for an organization's identification and update of digital opportunities and resources. The composition of this digital capability can serve as the starting point of future digital sustainability research, aiming at the research to determine the factors promoting the process of digital sustainable entrepreneurship of SMEs.

Next, we analyzed the mechanism of action of DIO in DSE from the angle of digital capability and, with DIO as the mediating variable, built the DC–DIO–DSE research framework for explaining by what means the enterprises possessing digital capability reconcile some contradictory phenomena between digitalization and sustainable development. This study confirmed that DC can positively affect DIO, which coincides with the conclusion of the empirical research by Khin and Ho (2018) [81]. We also found that enterprises can successfully unleash digital capability via digital innovation orientation to produce new sustainable products, processes, and services with the support of digital technology [73] and reduce the conflicts between digital energy consumption and social and environmental value. Our research conclusion extended the theoretical space of enterprise competence under the background of digitalization, which is conducive to deepening the knowledge about the mechanism of action of how DC affects DSE, as an extension to the theory of digital sustainable entrepreneurship proposed by Baranauskas and Raišienė (2022) [38].

Finally, our study provided empirical evidence for realizing the importance of a manager's cognition of sustainability guiding DC and DIO to convert into DSE within SMEs. The empirical studies by previous scholars on the manager's cognition are limited mainly to the exploration of employee attitude and behavior [27,82]; some scholars have tested the impact of the manager's cognition on the strategic results at the enterprise level [8,42]. It is imperative to explore the bearing of the manager's cognition of sustainability on the outcome variables of corporate digital strategies. Digital innovation, as a business goal, tends to fluctuate with the changes in the market and business cycle, while our study found that the manager can keep the enterprise with DIO and unleash the DC advantage to serve DSE practices when ascertaining the rationality of strategies on the demands for social and environmental opportunities. This research finding extended the research of the cognitive theory with the outcome variables of sustainable strategies.

### 7.2. Managerial Implications

This paper provided the below practical insights for digital sustainable entrepreneurship of pollutive SMEs. First, our study responded to the appeal in the times of digital sustainability and proposed the idea of constructing digital capability against how to change and promote people's actions in sustainable entrepreneurship [83]. Actors and organizations from SMES can use our research results as a design basis to construct a system of digital capability including digital technology capability and digital dynamic capability. Therefore, SMEs and governments shall create advantages for an atmosphere where digital capability blends in with sustainability and advocate implementing new-type digital technologies and flexible policies to cope with social and environmental challenges. Moreover, certain contradictions seem to exist between digital capability and the pursuit for sustainability, since the development and application of digitalization technologies are typically accompanied with the generation of energy dissipation and lift the requirement on the rationalization of resource allocation. The study in this paper suggests selecting the low-energy consumption, long-term empowered digital technologies through repeated

screening and recombination of digital technologies to help SMEs construct sustainable digital capability. Next, managers shall be aware that the digital sustainability undertaking is imperfect and in dynamic development and need to cope with various emergent problems; only through continuous digital innovation can social and environmental value be created persistently. During digital innovation, pollutive SMEs should carry out relevant activities according to their level of digital capability; digital capability limits the diffusion of digital innovation, so combining the strategic thinking of sustainability would be more beneficial for the value of digital innovation. Finally, the level of the manager's cognition of sustainable opportunities shall be raised. When considering a sustainability issue as an opportunity for, rather than a threat to, corporate development, the manager is more likely to take forward-looking environmental strategies, accelerate the digital transformation, and serve the capability for undertaking sustainability activity.

### 7.3. Limitations and Future Research

First, in the aspect of research design, we mainly adopted cross-section data which might be less reliable than the causal conclusion drawn from the longitudinal study when the causal relationship between DC and DSE was explained. Future research can consider expanding around this causal relationship from the perspective of longitudinal design. Next, due to the particularity of the research object and the limitations of the author's time and funding, the sample size of this paper is relatively small and the research industry is mainly concentrated in the pollutive industry. Future research can consider longitudinal time series research. To more accurately discuss DSE, future research should focus on a larger area or scope, in more non-polluting industries, expand the sample size of the study, and control the impact of industries on corporate environmental strategies. Finally, some difference exists between the cognitive structure and cognitive process of the Top Management Team (TMT) and manager's personal cognition [55]. This study used the individual level to reflect the overall situation of the organization; future research can collect multiple individual samples and aggregate them into organizational level variables, for example, to select the sustainable cognition of TMT as the research object to more comprehensively examine the relationship between cognition and sustainable strategy option.

**Author Contributions:** Conceptualization, G.X. and J.Z.; data curation, G.H.; formal analysis, G.X.; funding acquisition, G.H.; investigation: G.X., J.Z. and G.H.; methodology, G.X. and J.Z.; project administration, G.X. and J.Z.; writing—original draft: G.X. and J.Z.; writing—review and editing: G.X., J.Z. and G.H. All authors have read and agreed to the published version of the manuscript.

**Funding:** This research was funded by Changchun University of Science and Technology Base Support Special Project (KYC-JDY-2021-02); Jilin Provincial Department of Education Scientific Research Project (JJKH20220783SK).

**Informed Consent Statement:** Informed consent was obtained from all subjects involved in the study.

**Data Availability Statement:** Some or all data and models that support the findings of this study are available from the corresponding author upon reasonable request.

**Conflicts of Interest:** The authors declare no conflict of interest.

## Appendix A

**Table A1.** Measurement scales.

| Constructs | Item Description | Source |
|---|---|---|
| DSE | DSE1. Strip several businesses that go against the integration of digital technology with social and environmental value creation<br>DSE2. Change the competition approach of business departments, to achieve the goal of digital capability supporting social and environmental value creation<br>DSE3. Launch multiple digital transformation programs that can boost the productivity of social and environmental value creation of business departments<br>DSE4. The company is utilizing digital technology to improve its profitability for a better financial condition under sustainability<br>DSE5. The company usually uses digital technology to improve customer loyalty and image<br>DSE6. The company usually utilizes digital technology and take measures beneficial to the environmental and resource protection | Shepherd and Patzelt (2011) [62]; Baranauskas and Raišienė (2022) [38] |
| DC | DC-A1. Acquiring important digital technologies<br>DC-A2. We can interact digitally in activities for our employees, such as training, coprocessing, etc.<br>DC-A3. Manage digital technology and give full play to the functions brought by digital technology (e.g., analytical, network, connection, visualization, intelligence, etc.)<br>DC-B1. Identifying new digital opportunities<br>DC-B2. We are constantly searching for technological trends<br>DC-B3. We are able to analyze the signals scouted and analyze the digital scenarios of the future<br>DC-B4. We build a long-term digital vision and digital thinking for the company<br>DC-B5. We are able to reallocate resources quickly<br>DC-B6. We allow for repositioning and change<br>DC-B7. We have the ability to generate new ecosystems<br>DC-B8. We are able to leverage digital knowledge from within the organization | Zhou et al. (2010) [63]; Annarelli et al. (2021) [40] |
| DIO | DIO-A1. We use digital technology to improve corporate efficiency (including production efficiency, R&D efficiency, and communication efficiency)<br>DIO-A2. We use digital technology to reduce costs as a goal (including all aspects of R&D, production, and sales)<br>DIO-A3. We use digital technology to reduce information misalignment as a goal (e.g., by building digital platforms to enable data sharing)<br>DIO-B1. We use digital technology to deliver new offerings as a goal<br>DIO-B2. We use digital technology to introduce new players into the value creation process (e.g., building personalized platforms and using customer knowledge to create value)<br>DIO-B3. We use digital technology to adopt new ways of working<br>DIO-C1. We use digital technology to make organization structures more reasonable<br>DIO-C2. We use digital technologies to develop more advantageous business models<br>DIO-C3. We use digital technology to create new connections with stakeholders and collaborate on value creation | Von (2018) [64] |
| MCSO | MCSO1. I describe the overall natural and social environments confronting the company as opportunities for it<br>MCSO2. For the development of this company, I think the confronting natural and social environments are positive<br>MCSO3. I have perceived the amazing promotion of natural and social environment conditions for the future of the company<br>MCSO4. I think the natural and social environments confronting the company are controllable | Whit et al. (2003) [65]; Liu et al. (2013) [66] |

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
