# Peer review of "Digital Sustainable Entrepreneurship: A Digital Capability Perspective through Digital Innovation Orientation for Social and Environmental Value Creation"

_sustainability, doi:10.3390/su141811222_

Round 1

Reviewer 1 Report

Dear authors,

Many congratulations on your excellent article.

There are only two requests I would like to see fulfilled as a reviewer:

1. Put the output of the estimated model in AMOS in the results, so

After table 7 or after the graphs.

Tables alone do not give the full robustness of statistical legitimacy to the article.

2. In the abstract you should clearly specify that you have used a structural equation model using confirmatory factor analysis to estimate the cause and effect model. And that they used the software AMOS (I see that they use Amos 20, but I am sure they know that Amos 27 already exists. It would be good if in future studies they used the most recent software).

Best Regards 

Author Response

  1. Put the output of the estimated model in AMOS in the results, soAfter table 7 or after the graphs.

Response (1):

Thank you very much for your positive comments. We appreciate the time and effort you have put into our manuscript very much. According to your very careful comments, we have revealed estimated model in AMOS after table 6.

  1. In the abstract you should clearly specify that you have used a structural equation model using confirmatory factor analysis to estimate the cause and effect model. And that they used the software AMOS.

 Response (2):

Thank you very much for your valuable comments. Following your suggestions, we supplement the abstract, and the corresponding parts in the revised version are marked in red font.

Thanks again for your suggestion on the version of AMOS used, AMOS has launched AMOS28. We should update it too. We purchased AMOS.27 and re-analyzed the data with consistent results.

Reviewer 2 Report

Great article! Congrats!

Author Response

Thank you very much for your positive comments. We appreciate the time and effort you have put into our manuscript very much. Wish all the best wishes for you.

Reviewer 3 Report

Dear Authors,

I really enjoyed reading and reviewing your paper. While the paper has significant merits, there are some concerns that require further improvement.

1. The scope of the paper drifts throughout the paper. The paper rightly defines Sustainable entrepreneurship "as something focused on the preservation of nature, life support, and community in the pursuit of perceived opportunities to bring into existence future products, processes, and services, ...".  However, then the paper collects data about digitalization of SMEs in pollutive industries without actually supporting whether this digitalization can and should be considered as entrepreneurship. The digitalization of these firms may not sufficiently meet the definitions given in this paper, as well as in the literature for entrepreneurship. The authors need to support this claim explicitly showing how they meet this condition, or how they filter out responses related with regular digitalization. Why corporate digital transformation and digitalization should be accounted as entrepreneurship?

2. Inadequate presentation of the research model / question at the introduction. Authors presented the model to be tested at the end of the introduction section without enough support and reasoning in this section. I would recommend authors to give a simple research question and keep the presentation of the research model to the later sections.

3. Sampling needs to be better discussed. How the context and the sampling of data from pollutive industries in China from SMEs would impact the results of this study. How much the results are and can be generalizable. The authors need to discuss whether and how the selection of the sample would limit the results.

4. Discussion is premature. The results should be discussed against existing literature. In the current version, the discussion section is just two paragraphs. For the maturity of the paper, this section should be further elaborated. 

5. Claims and statements need referencing. A number of sentences in the introduction and literature review make a general statement about the state of the extant research without giving proper citations, or the number of citations are not enough as the prior statement suggests more than one research article. I advice authors to critically read the paper and whenever they make a claim, they need to support this with immediate citation. Otherwise, following the argumentation and decomposing what you believe and what you found in the literature may easily be confused by the reader.

6. Why a new construct. Why do we need to introduce Digital Innovation Orientation as a new construct is not agued convincingly. How they are different than IO or EO, and why digital version of an IO should be differentially defined?

7. Multilevel model and Level of analysis Issues. Cognition of managers are an individual level phenomena, while innovation orientation and capability are assumed as organizational variables/properties. The study collects data from individuals and considers their organizational averages or single informant responses (not clear) as organizational aggregates. In any case, level of analysis in the model, as well as the collected data should be better discussed. When there is a mismatch like cognition/orientation, than it needs to be theoretically discussed.

While not compulsory, I would also recommend authors to visit hybrid organizations literature as these topics are widely discussed in that line of research. Involving that literature would further enrich the article.

Also, I think capability-based view needs more citations for a stronger presentation.

I wish greatest success in your future publications.

Author Response

Point (1): The scope of the paper drifts throughout the paper. 

Thanks very much for your positive comments and hard work. We strongly agree with your comments concerning this issue. We believe that the addition of this part of the content will greatly improve the rationality of the theoretical logic of the paper. Following your suggestions,In the Introduction, Literature Review, Hypotheses and Discussion sections (The corresponding parts in the revised version are marked in red font), we supplement the discussion on why digital transformation or digitization of enterprises should be accounted as entrepreneurship, and the relationship between digitization and sustainability of SMEs in polluting industries, and provide relevant literature evidence.

First, we supplement and expound the definition of digital transformation in the literature review of digital sustainable entrepreneurship, and reveal the entrepreneurship contained in digitalization. Digitization of the enterprise brings unique changes in business operations, business processes and value creation(Libert et al., 2016).Digital transformation is defined as an organizational shift towards big data, business analytics, cloud computing, mobility and social media platforms(Nwankpa and, Roumani, 2016.).From the perspective of science and technology giving theory, digital transformation is a dual structure process composed of actors and digital technology(Volkoff and Strong, 2013).From the perspective of Sense-Making theory, digital transformation is a process of re-cognition and re-adaptation of strategy and structure based on digital technology(Zammuto,2007).Whether it is strategic or organizationally cognitive, the beginning of digitalization means the beginning of a new model.Digital transformation can subvert traditional industries under the development of digital opportunities. Its essence is the process of corporate entrepreneurship or strategic renewal(Warner and Wäger, 2019).Digital capability is one of the main outcomes of digital transformation(Proksch et al. , 2021).From the perspective of digital transformation, Nambisan (2017) pointed out that digital capabilities can be used to create new business models. Therefore, we believe that the more perfect the digital capability system is, the more active the corresponding entrepreneurial culture will be.

Second, to more strongly support the claim that digitization can and should be considered entrepreneurship, we add to our research hypotheses why digitization capabilities can be considered part of sustainable entrepreneurship(line220-230).We emphasized that the environmental and social supervision of small and medium-sized enterprises in China's polluting industries is the most stringent. In contrast, they lack the capital R&D and investment advantages of large enterprises, and they need to use digital intelligence and agility to respond to sustainable challenges.The study found that digital capabilities will facilitate the exchange of new ideas between SMEs and their value chain partners, which in turn will allow manufacturers to further improve or create business processes and related products.(Bharadwaj et al,2013)This means that digital capabilities will reduce unnecessary material and energy losses in the production process through lean management, reduce carbon emissions, and help carbon peak and carbon neutrality.(Yang et al,2010);and also help promote digital, networked and intelligent production and achieve green Energy-saving production,which is to realize social value(Yang et al,2011),In fact, this process is also a manifestation of digital innovation orientation (the mediating variable of this study). In addition, digital capabilities will accelerate the connection and integration of various entities in the enterprise value chain and their business activities, and then quickly realize personalized innovation to meet the individual needs of customers, that is, to achieve social value(Lenka et al,2017).

By complementing the above two aspects, we have reason to support the claim that digitization is seen as entrepreneurship

Point (2): Inadequate presentation of the research model / question at the introduction. 

Thank you very much for your positive comments. Following your suggestions,We have adjusted the description in the last paragraph of the Introduction. Removed the introduction about no research to test the model, and gave a simple research question to determine the composition of digital competence and its connection with digital sustainable entrepreneurship, and try to reveal the mechanism of this process and its influencing factors. Finally, we introduce the chapter arrangement of the thesis.(Line113-121)

Point (3):Sampling needs to be better discussed.

Thank you very much for your positive comments. We appreciate the time and effort you have put into our manuscript very much. According to your very careful comments, we made corresponding supplements in the original text(Line389-406)and explained it as follows:

China has many SMEs in polluting industries. According to statistics, there are 2.9241 million environmental protection related SMEs in China in 2020. SMEs in pollutive industries account for nearly half of all the number of SMEs. Sustainability has become a common management and management principle pursued by the vast majority of enterprises(Shepherd and Patzelt,2011).The research topic of this paper is digital sustainable entrepreneurship. Digitization is the premise of the sample selection for this study. The level of digital transformation activities of SMEs in pollutive industries varies significantly,and these SMEs have an important impact on the environment and society, and are more likely to be pressured by the government, the public, and the media to take environmental protection measures and implement sustainable entrepreneurship strategies.The sample SMEs in this study cover mining, papermaking, textile and other industries, which are representative to a certain extent. Selecting SMEs in this part of the industry as our research sample is more conducive to revealing the process of digital sustainable entrepreneurship from the perspective of digital capabilities.

We take pollutive SMEs with industry specificity as samples to examine the laws of digital sustainable entrepreneurial activities. Other SMEs in non-polluting industries with general characteristics are also faced with social and environmental externalities.These enterprises also carry out digital transformation to different degrees and have different digital capabilities. The conclusions of this study can also provide valuable reference for enterprises in other industries, but further tests need to be combined with empirical data. Therefore, we put this limitation in the part of Limitations and future research, and hope that other scholars can use the research results of this paper to carry out further tests.(Line756-762)

Point (4):Discussion is premature.

Thank you very much for your positive comments. Following your suggestions, We have adjusted it,We divided the discussion part into three paragraphs to discuss the conclusions of this study from three aspects: digital capability and digital sustainable entrepreneurship, the mediating role of digital innovation orientation, and the moderating role of managers' perception of sustainable opportunities.The discussion between the findings and the existing literature was strengthened.The corresponding parts in the revised version are marked in red font.

Point (5):Claims and statements need referencing.

Thank you very much for your positive comments. Following your suggestions, We have strengthened some citations in the Introduction and Literature Review sections to avoid readers' misunderstanding of the research perspective.The corresponding parts in the revised version are marked in red font. 

Added citations as follows:

[11]Nambisan S. Digital entrepreneurship: Toward a digital technology perspective of entrepreneurship. Entrep.Theory.Pract, 2017, 41, 1029-1055.

[12]Si, S.; Ahlstrom, D.; Wei, J.; Cullen, J. Business, entrepreneurship and innovation toward poverty reduction. Entrep.Region.Dev. 2020,32, 1-20.

[18]Ahmadova G, Delgado-Márquez B L, Pedauga L E, et al. Too good to be true: The inverted U-shaped relationship between home-country digitalization and environmental performance. Ecol Econ, 2022, 196: 107393.

[26]Lorenz, R.; Buess, P.; Macuvele, J.; Friedli, T.; Netland, T.H. Lean and Digitalization-Contradictions or Complements? In Proceedings of the Advances in Production Management Systems: Production Management for the Factory of the Future, Austin, TX, USA, 1–5 September 2019; pp. 77–84.

[27]Sabbir M M, Taufique K M R. Sustainable employee green behavior in the workplace: Integrating cognitive and non‐cognitive factors in corporate environmental policy. Bus.Strateg.Environ, 2022, 31,110-128.

[30]Libert B, Beck M, Wind Y. 7 Questions to ask before your next digital transformation. Harvard.Bus.Rev,2016, 60,11-13.

[31]Cozzolino A, Corbo L, Aversa P. Digital platform-based ecosystems: The evolution of collaboration and competition between incumbent producers and entrant platforms. J.Bus.Res, 2021, 126,385-400.

[32]Nwankpa, J.K.; Roumani, Y. IT Capability and Digital Transformation: A Firm Performance Perspective. In ICIS Proceedings; Association for Information Systems: Dublin, Ireland, 2016; pp. 1-16.

[33]Volkoff O, Strong D M. Critical realism and affordances: Theorizing IT-associated organizational change processes. MIS.quarterly, 2013, 819-834.

[34]Zammuto R.F., Griffith T.L., Majchrzak A., Dougherty D.J., Faraj S. Information technology and the changing fabric of organization. Organ.Sci, 2007, 18,749-762.

[36]Proksch D, Rosin A F, Stubner S, et al. The influence of a digital strategy on the digitalization of new ventures: The mediating effect of digital capabilities and a digital culture. J.Small.Bus.Manage, 2021: 1-29

[44]Bharadwaj A, El Sawy O A, Pavlou P A, et al. Digital business strategy: Toward a next generation of insights. MIS .Quarterly, 2013,37,471-482.

[51]Chege S M, Wang D, Suntu S L. Impact of information technology innovation on firm performance in Kenya. Inform.Technol .Dev, 2020, 26,316-345.

[74]Pan S L, Carter L, Tim Y, et al. Digital sustainability, climate change, and information systems solutions: Opportunities for future research. Int.J.Inform.Manage, 2022, 63,102444.

[76]Queiroz M M, Wamba S F. Blockchain adoption challenges in supply chain: An empirical investigation of the main drivers in India and the USA. Int.J.Inform.Manage, 2019, 46,70-82.

[77]Rajput S, Singh S P. Connecting circular economy and industry 4.0[J]. International Journal of Information Management, 2019, 49: 98-113.

[78]Kunkel S, Matthess M. Digital transformation and environmental sustainability in industry: Putting expectations in Asian and African policies into perspective. Environ.Sci.Policy, 2020, 112, 318-329.

[80]Whyte P, Lamberton G. Conceptualising sustainability using a cognitive mapping method. Sustainability, 2020, 12, 1977.

[83]Lichtenthaler U. Digitainability: the combined effects of the megatrends digitalization and sustainability. J Innov Manag, 2021, 9,64-80.

Point (6): Why a new construct.

Thank you very much for your positive comments. Following your suggestions, we explain as follows,and the corresponding parts in the revised version are marked in red font:

We supplemented the differences between the three in 3.2 Hypotheses about digital capabilities and digital innovation orientation and 3.3 Digital innovation orientation mediation, respectively, and emphasized the implications of introducing digital orientation for the entire model in the Discussion section.

Digital capability is not only a kind of application capability of digital technology (Nambisan, 2017, ), but also a dynamic capability of combining internal and external technologies, resources, opportunities and capabilities to adapt to changes in the digital environment (Shen et al,2021).Therefore, we divide digital capabilities into digital technical capabilities and digital dynamic capabilities. Digital capabilities are the result of digital transformation and are the foundation for all digital-related work(Proksch et al. , 2021).Digital innovation orientation represents the mechanism that involves the organization creative behavior for the implementation of new methods in order to solve the existing problems via the emerging technologies(Chege et al,2020;Shen et al,2021).Different from conventional innovation or innovation orientation, digital innovation breaks the process of traditional innovation and can use digital technology to carry out innovative experiments with low cost and high efficiency.We use three dimensions to measure the digital innovation orientation of enterprises, namely(1) efficiency (forming new processes; centered on production management and control, supply chain management, financial management, and control); (2) novelty (new products and services; R&D innovation and centered on customer service); and (3) convergence (forming new organizational structures and business models; centered on business management decisions).

Companies must find ways to innovate with relevant digital technologies by developing a digital transformation strategy(Hess et al,2016)That is to say, digital innovation orientation is a specific action process, method or path after possessing digital capabilities.Both digital technology capabilities and digital dynamic capabilities are formed for innovative development, and the ultimate goal of both must be to serve the strategic results of the enterprise(Yoo,et al. , 2012).Here we focus on the impact mechanism of digital capabilities on digital sustainable entrepreneurship. Digital sustainable entrepreneurship, as a sustainable action under digital transformation, is how to maintain the sustainable development of digital.The relationship between digitization and sustainability is still debated in previous studies (Ahmadova et al. , 2022),and our introduction of digital innovation can explain why digital capabilities can continue to promote digital sustainable entrepreneurship.This is because the digital innovation orientation can constantly try to renew and innovate business models to solve the development problems in digital and sustainable development. This discovery is also one of the main contributions of this study.

Point (7):Multilevel model and Level of analysis Issues.

Thanks again for your patient review.We have addressed and discussed the issue of multi-level models and levels of analysis at the beginning of the research design.Ideally, we should collect multiple managers and analyze the aggregation degree of the management team, confirming within-group consistency of individual respondents for each variable and take the average to reflect the overall organizational awareness level. But limited by our research funding and implementation difficulty, we did not take this approach. However, we have used the methods used in other literatures for reference and discussed theoretically with the context of Chinese SMEs.

We believe that choosing a senior or middle manager can also reflect the overall cognitive level of corporate managers. This is because in organizational research, the individual level is mainly from the top managers of the enterprise, especially the CEO's cognitive structure and the impact of cognitive processes on corporate strategic choices.Staw argues that TMT-level and organizational-level perceptions may be included in CEO perceptions when the CEO has decisive, centralized power.(Staw,1991)Due to the large proportion of private enterprises in this study, and most of the executives of private enterprises are founders or family heirs, they have a high degree of power concentration and absolute control.

In addition, the formulation and implementation of corporate strategies are mainly realized in two forms: top-down and bottom-up. The so-called top-down means that the senior of the company formulates the company's strategy according to their own experience and the future development direction of the company. The strategy formulated by the top management is implemented through the middle managers; while the bottom-up strategy is mainly formulated based on the actual experience and working conditions of the middle managers of the enterprise, and reported to the senior managers to form the final strategy of the enterprise.Middle managers play an important role in the formulation and implementation of corporate strategies, and can influence the formulation and implementation of corporate strategies by influencing the initial behavior and cognitive changes of subordinates.Therefore, the research object of this paper is defined as the senior managers or middle managers of various enterprises, such as the person in charge of environmental affairs, while ensuring that the selected middle and senior managers play an important role. They can influence the decision-making process for the formulation, selection and implementation of corporate environmental strategies.Therefore, it is also reasonable for us to use one senior or middle manager to reflect the overall organizational situation.We have supplemented this in the Sample and data section.(Line389-408).

Of course, we also encourage other scholars to use aggregated or team-level data to conduct research in the future, for which we have supplemented this in the Limitations and future research section.(Line768-772).

Finally, we also supplement the literature based on the ability-based perspective, which has been marked with red fonts in the text.

Thank you very much for your careful review again!

Round 2

Reviewer 3 Report

Thank you for the response. I am satisfied by the improvements in the paper and your explanations. Best regards,